# The role of inoculum dispersal and plant species identity in the assembly of leaf endophytic fungal communities

**Kevin D. Ricks**[¤], **Roger T. Koide**[ID]*

Department of Biology, Brigham Young University, Provo, UT, United States of America

¤ Current address: Program in Ecology, Evolution and Conservation Biology, University of Illinois at Urbana-Champaign, Urbana, IL, United States of America
* rogerkoide@byu.edu

**Data Availability Statement:** FASTQ files are deposited in the Sequence Read Archive (SRA) at NCBI (accession number PRJNA518913).

**Funding:** Funding to RTK was provided by Brigham Young University, The Charles Redd Center for

## Abstract

Because of disturbance and plant species loss at the local level, many arid ecosystems in the western USA benefit from revegetation. There is a growing interest in improving revegetation success by purposefully inoculating revegetation plants with mutualistic endophytic fungi that increase plant stress tolerance. However, inoculant fungi must compete against fungi that indigenous to the habitat, many of which may not be mutualistic. Our overall goal, therefore, is to learn how to efficiently colonize revegetation plants using endophytic fungal inoculum. The goal will be facilitated by understanding the factors that limit colonization of plants by endophytic fungi, including inoculum dispersal and host compatibility. We analyzed endophytic fungal communities in leaves of *Bromus tectorum* and *Elymus elymoides* (Poaceae), *Chrysothamnus depressus* and *Artemisia tridentata* (Asteraceae), *Alyssum alyssoides* (Brassicaceae) and *Atriplex canescens* (Amaranthaceae), each occurring in each of 18 field plots. We found that dispersal limitation was significant for endophytic fungal communities of *Atriplex canescens* and *Bromus tectorum*, accounting for 9 and 17%, respectively, of the variation in endophytic fungal community structure, even though the maximum distance between plots was only 350 m. Plant species identity accounted for 33% of the variation in endophytic fungal community structure. These results indicate that the communities of endophytic fungi assembling in these plant species depend significantly on proximity to inoculum source as well as the identity of the plant species. Therefore, if endophytic fungi are to be used to facilitate revegetation by these plant species, land managers may find it profitable to consider both the proximity of inoculum to revegetation plants and the suitability of the inoculum to targeted host plant species.

## Introduction

Revegetation may become a management necessity when severe habitat disturbance leads to a significant loss in plant cover. This is certainly the case in the arid, western region of the USA, where revegetation is becoming increasingly necessary as a consequence of native species

Western Studies, The Roger and Victoria Sant Endowment for a Sustainable Environment, and the Sustainable Bioenergy Research Program of the USDA National Institute of Food and Agriculture (# 2011-67009-20072).

**Competing interests:** The authors have declared that no competing interests exist.

losses due to land use change [1] and increased fire frequency following invasion by non-native grasses [2–5]. However, revegetation may be difficult in the physically stressful habitats characteristic of the arid, western USA [6], which suffer from water stress and extreme temperatures [7].

Endophytic fungi have been found in all plant species investigated thus far [8–10]. They frequently form complex communities within plant tissues comprising dozens of species [8,11,12]. In some cases, the vigor of the plant is significantly improved as a consequence of colonization by these fungi, especially under stressful conditions. For example, endophytic fungi may increase plant resistance to herbivory [13,14] and disease [9,15,16], and tolerance to heat [16–19], cold [20] and drought [16,19–21]. It is not surprising, therefore, that there is significant interest in utilizing mutualistic, endophytic fungi to facilitate revegetation, particularly in stressful habitats [22–25].

While it is appealing to inoculate revegetation plants with beneficial endophytic fungi, such an approach may not be effective if inoculant taxa do not compete well against indigenous endophytic fungi, especially because many indigenous taxa will not be particularly beneficial to their plant hosts. Depending on the combination of fungal taxon, plant taxon and environment, endophytic fungi range from mutualistic [8,18,26] to latent pathogenic [27,28] and latent saprotrophic [29,30]. Because desirable inoculant endophytic fungi will have to compete with indigenous endophytic fungi, effective and low-cost inoculation strategies will require understanding and overcoming the major constraints to colonization of plants by inoculant strains.

Two of the potentially important factors that influence the colonization of plant tissues by endophytic fungi are dispersal limitation from inoculum source to target plants, and compatibility of inoculant fungi with the target plant species. Significant dispersal limitation [31] and biogeographical pattern in the distribution of endophytic fungi [12] suggest that proximity to a source of inoculum influences the probability of a fungal taxon colonizing plant tissue. In addition, colonization of a particular host plant by a particular endophytic fungus is limited by the degree of compatibility between fungus and plant [32], so a given source of inoculum may produce significantly different communities of endophytic fungi depending on the plant species [11,33] or plant genotype [34–37].

In this study our goal was to characterize the impacts of both plant species identity and fungus dispersal-limitation on endophytic fungal community structure in leaves of six plant species common in the eastern Great Basin of the USA. In some previous studies, the distinction between plant species identity and dispersal-limitation could not be made because plant species identity was confounded by spatial location in the environment [12,38]. In order to distinguish between plant species identity and spatial location, we sampled six plant species in each of 18 small field plots to greatly reduce dispersal limitation among plant species within plots in order to quantify dispersal limitation among plots.

## Materials and methods

The vegetation at our study site (40˚5'34.7" N, 112˚19'37.2" W, approximately 10 km east of Vernon, UT) is sagebrush-steppe. The site is administered by the United States Department of the Interior, Bureau of Land Management. No specific permissions were required because no plants were removed from the site and the study did not involve endangered or protected species.

Common plant species at the site include *Bromus tectorum* L. and *Elymus elymoides* Raf., mycorrhizal members of the Poaceae, *Artemisia tridentata* Nutt. subsp. *wyomingensis* Beetle & Young and *Chrysothamnus depressus* Nutt., mycorrhizal members of the Asteraceae, and

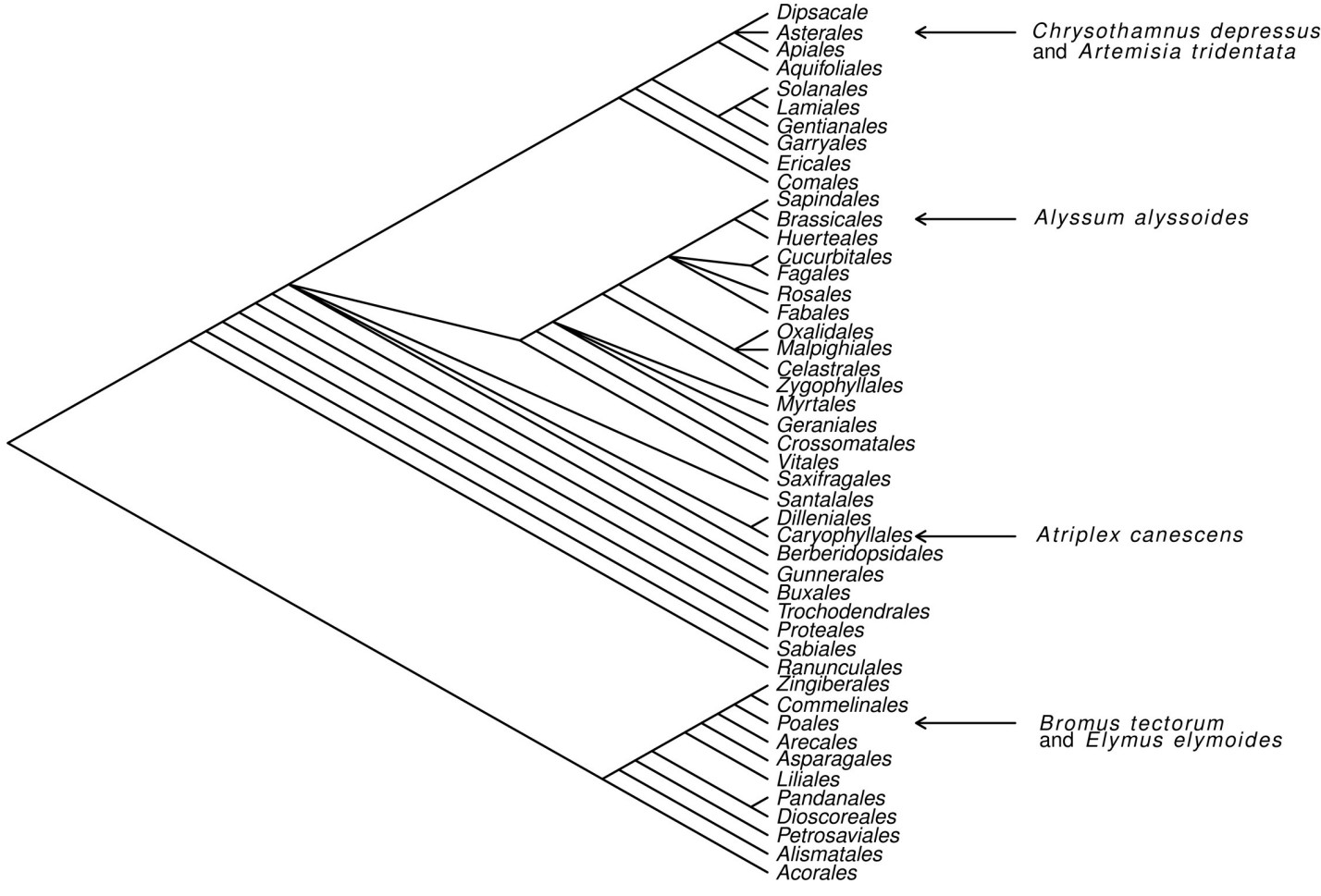

**Fig 1. Phylogeny of major angiosperm orders with the locations of the six plant species sampled in this study, redrawn from Bliss *et al.* [39].**

*Atriplex canescens* Nutt. and *Alyssum alyssoides* L., nonmycorrhizal members of the Amaranthaceae and Brassicaceae, respectively (Fig 1).

Within the study site, we designated 18 plots (each approximately 16 m$^2$) as locations where individuals of all six plant species listed above were present within an approximately 5 m radius. The total area encompassing all 18 plots was approximately 0.5 km$^2$, and the maximum distance between plots was approximately 350 m. Eighteen replicate plots were established to provide sufficient statistical power to accurately characterize foliar endophytic fungal communities for each of the plant species [30]. From each plot we randomly sampled 5 disease-free leaves from one randomly-selected individual of each of the six species on 15 May 2017. All sampled leaves were placed on ice in the field. Upon returning to the laboratory later in the day, samples were stored temporarily at 6˚C. The 5 leaves from each species in a plot were pooled into a single species sample. There were, therefore, a total of 108 samples (6 species x 18 plots).

All samples were processed within 5 days of collection. To destroy external (epiphytic) fungal DNA from leaf samples, we placed the samples in 3% sodium hypochlorite (NaClO) and 1% Tween-20 for 20 minutes, after which they were rinsed thoroughly in sterile water [40–42]. We tested this surface preparation method independently on corn and oak leaves and found it to be effective in eliminating leaf surface fungal DNA (Clark & Koide, unpublished data).

Approximately 0.5 g of plant tissue from each pooled sample were placed in Mo Bio Power-plant Pro DNA extraction tubes (Mo Bio Laboratories Inc., Carlsbad, CA, USA), and DNA was extracted following the standard protocol for the extraction kits with one exception. Instead of agitating the sample on a standard benchtop vortex mixer with the Mo Bio Vortex Adaptor, we agitated with a 2010 Geno/Grinder (SPEX SamplePrep, Metuchen, NJ, USA) at 1000 rpm for 4 min. All extracted DNA samples were stored at −20˚C until PCR amplification.

Samples were prepared for high-throughput sequencing using a two-step PCR amplification. In the first step, the ITS2 sub-region from the fungal ITS region was amplified using 5.8S Fun and ITS4 Fun primers [43]. The first PCR program was: hot-start activation at 95˚C for 15 min, 27 cycles of 95˚C for 30 s, 58˚C for 30 s, and 72˚C for 2 min with final elongation at 72˚C for 10 min. In the second PCR, barcodes and Illumina flowcell adapters were appended to the PCR1 amplicons. The second thermal cycling program was: hot-start activation at 95˚C for 15 min, 12 cycles of 95˚C for 30 s, 55˚C for 40 s, and 72˚C for 40 s with final elongation at 72˚C for 10 min. We used Apex Hot start PCR Master Mix (Apex Bioresearch Products, North Liberty, IA, USA).

Identical volumes of PCR2 product from each sample were pooled together to create the sequence library prior to sequencing. Sequencing was performed at the Institute for Bioinformatics and Evolutionary Studies (iBEST) genomics resources core at the University of Idaho (http://www.ibest.uidaho.edu/, Moscow, ID). Amplicon libraries were sequenced using 2 × 300 paired-end reads on an Illumina MiSeq sequencing v3 (600 cycles) platform (Illumina Inc., San Diego, CA, USA). Prior to all sequence analyses, we rarefied the samples to an equal sequencing depth (3,000 reads, see below) while maintaining adequate coverage (see accumulation curve, S1 Fig). This made samples comparable despite the potential for different original sequencing depths.

The initial bioinformatic processing occurred in the DADA2 pipeline [44]. We eliminated reads with quality scores less than 10 (truncQ = 10). Paired reads were assembled using the mergePairs function with a minimum overlap of 20 bp and allowing a maximum mismatch of 5% within the region of overlap. Non-overlapping reads were joined with a 10 bp sequence of Ns. Using the UCHIME function, 0.8% of reads were identified as chimeras, and these were removed. A small portion (6%) of our ITS sequences belonged to the host plants. These were filtered out prior to analysis. The remaining reads were clustered into OTUs based on a 97% similarity criterion [45–47], using *de novo* OTU picking in the QIIME pipeline [47]. Using the UNITE database [48] as a reference, OTUs were assigned taxonomy using a Ribosomal Database Project Naïve Bayesian Classifier algorithm [49] with kmer size of 8, and 50% bootstrap threshold required to assign taxonomy. To compare samples at equivalent sequencing depth, all samples were rarefied to the lowest number of reads observed in all samples, which was 3,000. FASTQ files are deposited in the Sequence Read Archive (SRA) at NCBI (accession number PRJNA518913).

To compare endophytic fungal communities among plant species, we performed permutational multivariate analyses of variance (PERMANOVA) in the R statistical environment [50] with the Vegan package [51] using Bray-Curtis dissimilarities [52]. To make specific comparisons between plant species, we performed pairwise PERMANOVAs and, to protect against false positives, we used Benjamini-Hochberg false discovery rate adjustments on all *P* values [53]. We additionally visualized variation in endophytic fungal community structure among plant species using non-metric multidimensional scaling (NMDS) ordination employing Bray-Curtis dissimilarities, 25 perturbations and three axes, and displayed the ordinations using the first two axes.

We tested correlations between phylogenetic distances among plant species and dissimilarities in the structure of endophytic fungal communities among plant species with a Mantel test

in the vegan package [51] in R [50]. Phylogenetic distances among species were estimated with TimeTree [54], which estimates divergence time between species pairs from published phylogenetic trees. Dissimilarities in the structure of endophytic fungal communities among plant species were calculated as Bray-Curtis distances between the centroids of the replicate endophytic fungal communities of each plant species. We generated standard errors of these distances using bootstrap resampling [55]. If plant phylogeny contributes to the structure of endophytic fungal communities, the pairwise phylogenetic distances among plant species should be positively correlated with endophytic fungal community dissimilarities among plant species.

In order to determine whether the frequencies of occurrence for the various endophytic fungal OTUs varied among plant species or among plant families, we used chi-square analyses conducted in the R environment [50]. We protected against false positives by correcting all *P* values using the Benjamini-Hochberg false discovery rate adjustments [53].

To test whether dispersal of endophytic fungal inoculum limited the assembly of endophytic fungal communities, we used one-tailed Mantel tests in the vegan package [51] in R [50] for each of the six plant species, regressing geographic distances among plots against Bray-Curtis dissimilarities in community structure. Distances among plots varied from 20 to 350 meters. Dispersal limitation is evidenced by a significant correlation between geographical distance and fungal community structure.

## Results

Before quality filtering, there were an average of 12,816 reads per sample. After quality filtering, there were an average of 7,167 read per sample (688,091 reads in 96 samples, comprising 16 plots with a complete set of six plant species out of the original 18 plots). After rarefying to 3,000 reads in each sample, there were 394 unique endophytic fungal OTUs (see accumulation curves S1 Fig).

According to the Mantel tests, geographic distance among plots was significantly correlated with endophytic fungal community dissimilarity (Bray-Curtis distances) for *Atriplex canescens* and *Bromus tectorum*, accounting for 9 and 17%, respectively, of total variation in the structure of endophytic fungal communities (Table 1). For the other four plant species, there were no significant correlations. Thus, dispersal of endophytic inoculum over distances of a maximum of 350 m apparently constrained the structure of endophytic fungal communities in *Bromus tectorum* and *Atriplex canescens* but not in the other plant species.

Plant species identity was also significant with respect to the structure of endophytic fungal communities. According to the PERMANOVA, plant species identity was significant, accounting for 33% of the total variation in the structure of endophytic fungal communities ($R^2$ = 0.330, Table 2). This can be visualized in the corresponding ordination (Fig 2) and in the composition of the fungal communities (by fungal order) in each of the plant species (Fig 3).

**Table 1. Results of Mantel tests regressing geographic distances among plots against endophytic fungal community dissimilarity (Bray-Curtis distance).**

| Species | R | $R^2$ | *P* |
|---|---|---|---|
| *Bromus tectorum* | 0.414 | 0.171 | **0.001** |
| *Chrysothamnus depressus* | -0.042 | 0.001 | 0.495 |
| *Artemisia tridentata* | -0.268 | 0.072 | 0.988 |
| *Elymus elymoides* | -0.293 | 0.085 | 0.981 |
| *Atriplex canescens* | 0.303 | 0.092 | **0.025** |
| *Alyssum alyssoides* | -0.185 | 0.034 | 0.923 |

There were 18 fungal taxa that were capable of colonizing leaves of all six of the plant species, although not at the same frequency (Table 3). These taxa included Unknown Pleosporales 1, Unknown Lecanorales 2, *Tetracladium sp. 1*, *Coprinopsis brunneofibrillosa*, Unknown Lecanorales 1, Unknown Pleosporales 2, *Saccharomyces paradoxus*, Unknown Phaeosphaeriaceae 1, *Neocamarosporium* sp.*1*, Unknown Pleosporales 10, *Limonomyces culmigenus*, Unknown Sporormiaceae, *Ramimonilia apicalis*, *Comoclathris spartii*, Unknown Pleosporales 3, *Marchandiomyces lignicola*, *Dioszegia* sp., and *Alternaria alternata*.

Following quality filtering and rarefaction, the maximum frequency was 16.

The FDR-protected chi-square tests indicated that, of the 394 total endophytic fungal OTUs, there were 68 OTUs that exhibited significantly different frequencies among plant species (Table 3), and 62 of these exhibited significantly different frequencies among plant families (Table 3). For example, some fungal taxa were most frequent among the Poaceae including *Clavispora lusitaniae* and Unknown Pleosporales 11. Some fungal taxa were most frequent among the Poaceae and Asteraceae including Unknown Pleosporales 1, *Tetracladium* sp. *1*, Unknown Lecanorales 1 and *Saccharomyces paradoxus*. Some fungal taxa were least frequently occurring in *Atriplex canescens* (Amaranthaceae) including Unknown Lecanorales 2, *Coprinopsis brunneofibrillosa*, Unknown Pleosporales 2 and Unknown Phaeosphaeriaceae 1.

According to the Mantel test, endophytic fungal community dissimilarity among plant species (Bray-Curtis distance) was significantly correlated with phylogenetic distance among plant species, accounting for nearly 29% of total variability ($P = 0.006$, $R^2 = 0.286$). Because phylogenetic distance was confounded by mycorrhizal status in this Mantel test, we performed a partial Mantel test using mycorrhizal status as a covariate. The partial Mantel test was also significant ($P = 0.006$), and it explained more variation ($R^2 = 0.569$) than the Mantel test, confirming that in our system plant phylogeny was an important factor structuring endophytic fungal communities.

According to the pairwise PERMANOVAs (Table 4), the structure of the endophytic fungal communities of the two members of the Poaceae, *Bromus tectorum* and *Elymus elymoides*, were not significantly different ($P = 0.367$) and, similarly, that the structure of the endophytic fungal communities of the two members of the Asteraceae, *Artemisia tridentata* and *Chrysothamnus depressus*, were not significantly different ($P = 0.299$). However, all comparisons between species of different plant families were significantly different (Table 4). These results are consistent with the Mantel tests and suggest that variation among endophytic fungal communities due to plant species was mainly due to variation among plant families and not to variation among species within plant families.

Plant mycorrhizal status also appeared to contribute to variation in endophytic fungal community structure. The endophytic fungal communities associated with *Atriplex canescens* (nonmycorrhizal, Amaranthaceae) and *Alyssum alyssoides* (nonmycorrhizal, Brassicaceae) were more dissimilar from those of the four mycorrhizal plant species than the mycorrhizal plant species were to each other (Fig 2). This is also apparent from the Bray-Curtis distances, D (Table 4), which represent dissimilarities among fungal communities. These dissimilarities were greater between fungal communities of *Atriplex canescens* and those of the four

**Table 2. Results of PERMANOVA for fungal communities in leaves of the six plant species.**

| Factor | df | SS | MS | F | $R^2$ | $P$ |
|--------|-----|--------|-------|-------|-------|--------|
| Plant species | 5 | 6.582 | 1.316 | 8.799 | 0.330 | <0.001 |
| Plot | 15 | 2.163 | 0.144 | 0.964 | 0.108 | 0.626 |
| Residuals | 75 | 11.220 | 0.150 | | 0.562 | |
| Total | 95 | 19.965 | | | 1.000 | |

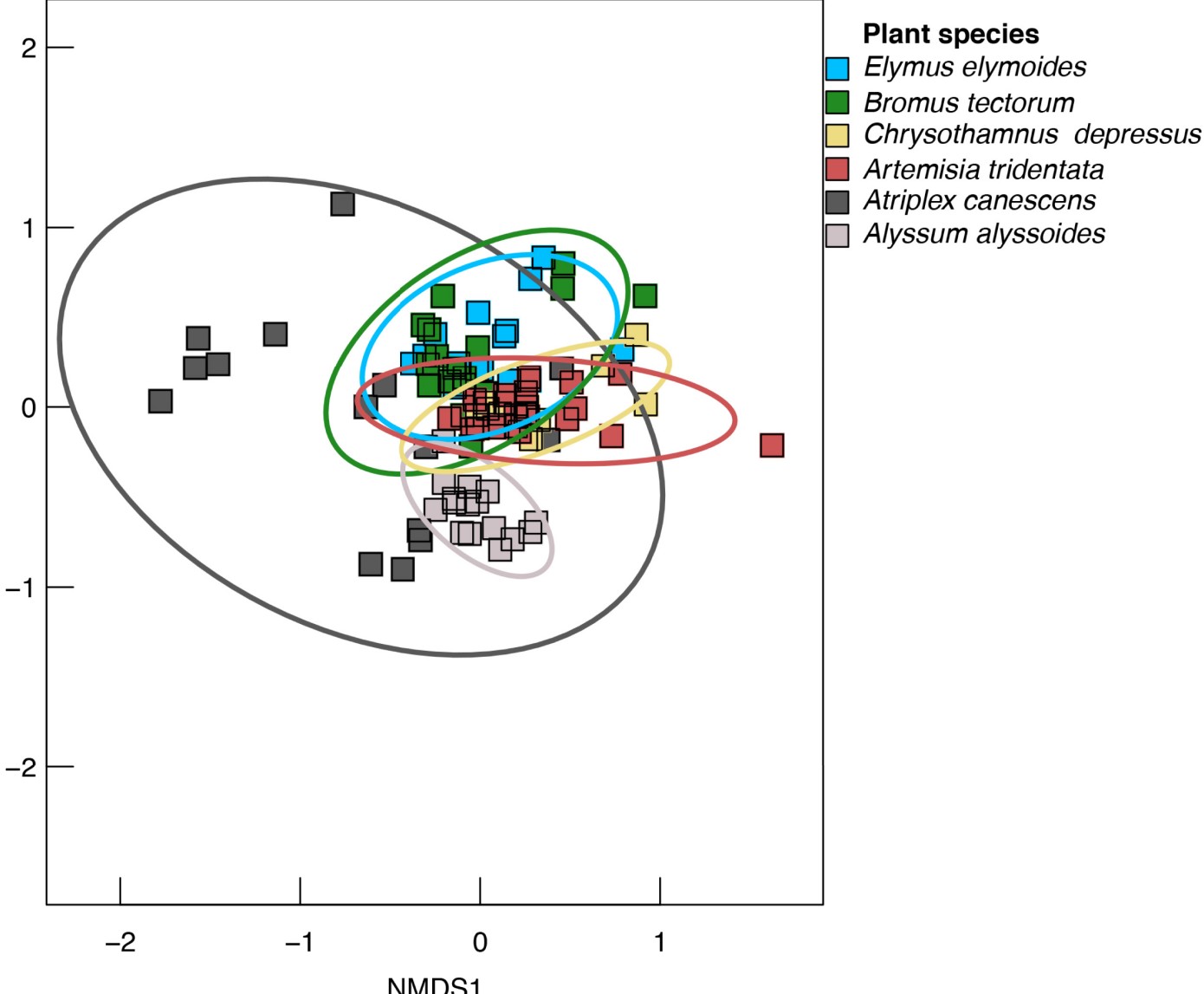

**Fig 2. NMDS ordinations visualizing the leaf endophytic fungal communities from the six plant species.** Ellipses are drawn to include 95% of the variation for each group. Stress = 0.137.

mycorrhizal plant species (range 0.477–0.510) and between those of *Alyssum alyssoides* and the four mycorrhizal plant species (range 0.513–0.616), than they were among the mycorrhizal plant species (range 0.221–0.362), despite the fact that *Atriplex canescens* and *Alyssum alyssoides* are more closely related to the Asteraceae than the Poaceae is to the Asteraceae. The endophytic fungal community of *Alyssum alyssoides* was also significantly different from that of *Atriplex canescens* (Table 4, Fig 2).

Some fungal taxa were most frequently occurring or uniquely occurring in *Atriplex canescens* (Amaranthaceae, Table 3) including *Powellomyces* sp. *1*, *Powellomyces* sp. *2*, Unknown Pleosporales 5, Unknown Pleosporales 7, Unknown Pleosporales 8, *Neocamarosporium* sp. *1*, *Naganishia cerealis*, *Sporomiella leporina*, *Dioszegia* sp., *Megaspora cretacea*, Unknown

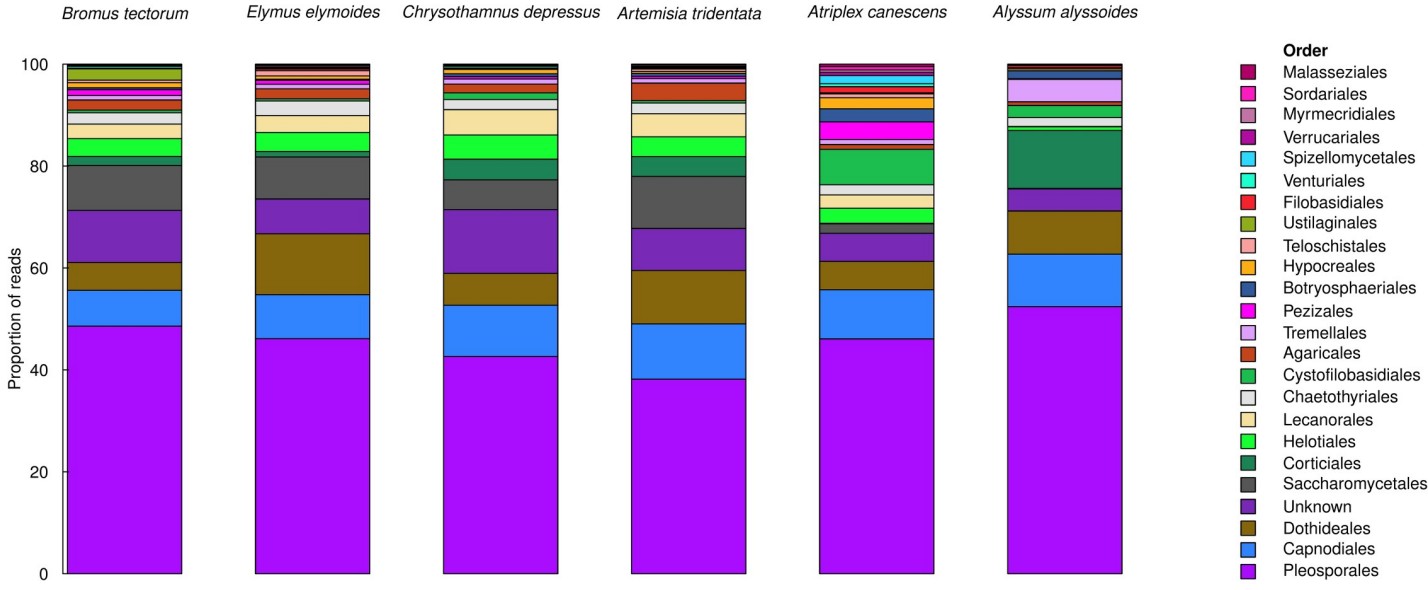

**Fig 3. Composition of the fungal comunities (by fungal order) in each of the plant species.** Data are proportional rarefied reads.

Phaeosphaeriaceae 3, Unknown Verrucariaceae, *Chaetosphaeronema* sp., Unknown Agaricales, *Malassezia globosa*, Unknown Lophiostomataceae, *Protrudomyces lateralis*, *Gibberella tricincta*, *Schizothecium carpinicola*, Unknown Melanommataceae, *Peziza sp*., *Coniosporium apollinis*, *Circinaria hispida*, Unknown Ustilaginales, *Filobasidium stepposum*, *Rhizophlyctis rosea*, *Spizellomyces sp*., *Kochiomyces* sp., *and Mortierella alpina*. Moreover, some fungal taxa were most frequently occurring or uniquely occurring in *Alyssum alyssoides* (Brassicaceae, Table 3) including Unknown Pleosporales 3, Unknown Pleosporales 4, Unknown Pleosporales 5, Unknown Pleosporales 6, *Unknown Pleosporales 9*, Unknown Pleosporales 10, Unknown Pleosporales 11, Unknown Pleosporales 12, Unknown Pleosporales 13, *Limonomyces culmigenus*, Unknown Pezizaceae, Unknown Sporormiaceae, *Ramimonilia apicalis*, *Comoclathris spartii*, *Marchandiomyces lignicola*, *Dioszegia hungarica*, *Naganishia cerealis*, *Neocamarosporium* sp. *2*, *Dioszegia* sp., *Filobasidium magnum*, *Sporormiella leporine*, Unknown Phaeosphaeriaceae 2, *Septoriella hirta*, *Dioszegia* sp., *Alternaria alternata*, *Comoclathris sedi*, *Vishniacozyma dimennae*, Unknown Agaricomycetes, *Phoma aloes*, *Phaeococcomyces mexicanus* and *Cyrenella elegans*.

## Discussion

Colonization of plants by mutualistic, endophytic fungi may have large, positive effects on plant tolerance to environmental stresses [17,20], which are commonly experienced in many parts of the arid, western region of the USA [2,3]. Therefore, revegetation success in that region may be increased when plants are colonized by these mutualistic fungi [56]. However, deliberate inoculation of plants with mutualistic fungi may not be as effective as hoped for in a field setting where natural sources of inoculum also contribute to the endophytic fungal communities, given the fact that not all such endophytic fungi are beneficial to plants [27,28,30,57]. To maximize the effectiveness of inoculant fungi, it is important to understand the constraints to their use, including the constraints imposed by inoculum dispersal limitation and host compatibility.

**Table 3.  Frequencies of occurrence for leaf endophytic fungal taxa exhibiting significant chi-square results.**

| OTU | Poaceae | | Asteraceae | | Amaranthaceae | Brassicaceae | Significantly different among families | Significantly different among species |
|---|---|---|---|---|---|---|---|---|
| | *Bromus tectorum* | *Elymus elymoides* | *Chrysothamnus depressus* | *Artemisia tridentata* | *Atriplex canescens* | *Alyssum alyssoides* | | |
| Unknown Pleosporales 1 | 15 | 15 | 13 | 14 | 7 | 6 | X | X |
| Unknown Lecanorales 2 | 14 | 15 | 12 | 13 | 8 | 12 | X | |
| *Clavispora lusitaniae* | 14 | 15 | 3 | 5 | 1 | 0 | X | X |
| *Tetracladium* sp. *1* | 14 | 12 | 10 | 12 | 8 | 5 | X | X |
| *Coprinopsis brunneofibrillosa* | 13 | 14 | 10 | 11 | 3 | 9 | X | X |
| Unknown Lecanorales 1 | 13 | 13 | 13 | 13 | 1 | 1 | X | X |
| Unknown Pleosporales 2 | 12 | 12 | 13 | 13 | 6 | 15 | X | X |
| *Saccharomyces paradoxus* | 12 | 11 | 11 | 14 | 6 | 3 | X | X |
| Unknown Phaeosphaeriaceae 1 | 10 | 13 | 10 | 15 | 3 | 15 | X | X |
| *Neocamarosporium* sp.*1* | 8 | 11 | 4 | 5 | 13 | 6 | X | X |
| Unknown Pleosporales 10 | 8 | 9 | 8 | 9 | 6 | 15 | X | |
| *Limonomyces culmigenus* | 8 | 8 | 11 | 9 | 2 | 15 | X | X |
| Unknown Pezizaceae | 8 | 8 | 4 | 3 | 8 | 0 | X | X |
| Unknown Sporormiaceae | 8 | 8 | 1 | 1 | 10 | 8 | X | X |
| *Ramimonilia apicalis* | 5 | 8 | 8 | 8 | 9 | 15 | X | X |
| Unknown Pleosporales 11 | 4 | 5 | 0 | 1 | 1 | 0 | X | |
| Unknown Pleosporales 12 | 4 | 0 | 0 | 0 | 0 | 0 | | X |
| *Comoclathris spartii* | 3 | 7 | 4 | 5 | 4 | 14 | X | X |
| Unknown Pleosporales 3 | 3 | 5 | 6 | 6 | 4 | 14 | X | X |
| *Marchandiomyces lignicola* | 3 | 4 | 1 | 1 | 2 | 10 | X | X |
| *Dioszegia hungarica* | 3 | 3 | 8 | 6 | 0 | 15 | X | X |
| *Naganishia cerealis* | 3 | 2 | 0 | 0 | 7 | 1 | X | X |
| Unknown Pleosporales 4 | 3 | 0 | 0 | 0 | 7 | 12 | X | X |
| Unknown Pleosporales 13 | 3 | 0 | 0 | 0 | 0 | 0 | | X |
| *Neocamarosporium* sp. *2* | 3 | 0 | 0 | 0 | 0 | 0 | | X |
| Dioszegia sp. | 2 | 3 | 3 | 2 | 2 | 10 | X | X |
| *Filobasidium magnum* | 2 | 1 | 0 | 2 | 6 | 8 | X | X |
| *Sporormiella leporina* | 2 | 1 | 0 | 0 | 6 | 0 | X | X |
| Unknown Phaeosphaeriaceae 2 | 1 | 5 | 5 | 7 | 0 | 14 | X | X |
| *Septoriella hirta* | 1 | 3 | 3 | 2 | 0 | 10 | X | X |

*(Continued)*

**Table 3.** (Continued)

| OTU | Poaceae | | Asteraceae | | Amaranthaceae | Brassicaceae | Significantly different among families | Significantly different among species |
|---|---|---|---|---|---|---|---|---|
| | *Bromus tectorum* | *Elymus elymoides* | *Chrysothamnus depressus* | *Artemisia tridentata* | *Atriplex canescens* | *Alyssum alyssoides* | | |
| *Dioszegia* sp. | 1 | 3 | 0 | 0 | 6 | 0 | X | X |
| *Alternaria alternata* | 1 | 2 | 2 | 2 | 3 | 12 | X | X |
| Unknown Pleosporales 5 | 1 | 2 | 1 | 3 | 7 | 0 | X | X |
| *Comoclathris sedi* | 1 | 2 | 0 | 3 | 7 | 11 | X | X |
| *Megaspora cretacea* | 1 | 0 | 0 | 1 | 5 | 0 | X | X |
| Unknown Phaeosphaeriaceae 3 | 1 | 0 | 0 | 0 | 4 | 0 | X | X |
| Unknown Phaeosphaeriaceae 4 | 0 | 4 | 0 | 0 | 0 | 0 | | X |
| Unknown Phaeosphaeriaceae 5 | 0 | 4 | 0 | 0 | 0 | 0 | | X |
| *Comoclathris spartii* | 0 | 4 | 0 | 0 | 0 | 0 | | X |
| Unknown Pleosporales 14 | 0 | 3 | 0 | 0 | 0 | 4 | | X |
| Unknown Lecanorales 3 | 0 | 3 | 0 | 0 | 0 | 0 | | X |
| *Tetracladium* sp. 2 | 0 | 3 | 0 | 0 | 0 | 0 | | X |
| Unknown Verrucariaceae | 0 | 1 | 0 | 0 | 5 | 0 | X | X |
| *Chaetosphaeronema* sp. | 0 | 0 | 0 | 1 | 4 | 0 | X | X |
| *Powellomyces* sp. 2 | 0 | 0 | 0 | 0 | 8 | 0 | X | X |
| Unknown Agaricales | 0 | 0 | 0 | 0 | 5 | 1 | X | X |
| *Malassezia globosa* | 0 | 0 | 0 | 0 | 5 | 1 | X | X |
| Unknown Lophiostomataceae | 0 | 0 | 0 | 0 | 4 | 0 | X | X |
| *Protrudomyces lateralis* | 0 | 0 | 0 | 0 | 4 | 0 | X | X |
| *Powellomyces* sp. 1 | 0 | 0 | 0 | 0 | 4 | 0 | X | X |
| *Gibberella tricincta* | 0 | 0 | 0 | 0 | 3 | 0 | X | X |
| *Schizothecium carpinicola* | 0 | 0 | 0 | 0 | 3 | 0 | X | X |
| Unknown Melanommataceae | 0 | 0 | 0 | 0 | 3 | 0 | X | X |
| *Peziza* sp. | 0 | 0 | 0 | 0 | 3 | 0 | X | X |
| Unknown Pleosporales 7 | 0 | 0 | 0 | 0 | 3 | 0 | X | X |
| *Coniosporium apollinis* | 0 | 0 | 0 | 0 | 3 | 0 | X | X |
| Unknown Pleosporales 8 | 0 | 0 | 0 | 0 | 3 | 0 | X | X |
| *Circinaria hispida* | 0 | 0 | 0 | 0 | 3 | 0 | X | X |
| Unknown Ustilaginales | 0 | 0 | 0 | 0 | 3 | 0 | X | X |
| *Filobasidium stepposum* | 0 | 0 | 0 | 0 | 3 | 0 | X | X |
| *Rhizophlyctis rosea* | 0 | 0 | 0 | 0 | 3 | 0 | X | X |
| *Spizellomyces* sp. | 0 | 0 | 0 | 0 | 3 | 0 | X | X |
| *Kochiomyces* sp. | 0 | 0 | 0 | 0 | 3 | 0 | X | X |

(*Continued*)

**Table 3.** (Continued)

| OTU | Poaceae | | Asteraceae | | Amaranthaceae | Brassicaceae | Significantly different among families | Significantly different among species |
|---|---|---|---|---|---|---|---|---|
| | *Bromus tectorum* | *Elymus elymoides* | *Chrysothamnus depressus* | *Artemisia tridentata* | *Atriplex canescens* | *Alyssum alyssoides* | | |
| *Mortierella alpina* | 0 | 0 | 0 | 0 | 3 | 0 | X | X |
| *Vishniacozyma dimennae* | 0 | 0 | 0 | 0 | 1 | 5 | X | X |
| Unknown Pleosporales 6 | 0 | 0 | 0 | 0 | 0 | 8 | X | X |
| Unknown Agaricomycetes | 0 | 0 | 0 | 0 | 0 | 5 | X | X |
| *Phoma aloes* | 0 | 0 | 0 | 0 | 0 | 4 | X | X |
| Unknown Pleosporales 9 | 0 | 0 | 0 | 0 | 0 | 3 | X | X |
| *Phaeococcomyces mexicanus* | 0 | 0 | 0 | 0 | 0 | 3 | X | X |
| *Cyrenella elegans* | 0 | 0 | 0 | 0 | 0 | 3 | X | X |

In some previous studies, it was impossible to separate the effects of plant species identity from dispersal limitation because plant species identity was confounded by geographic location [12,38]. Because we sampled all six plant species from each of the 18 small plots (approximately 16 m$^2$), each of the six plant species within a plot was presumably exposed to the same inoculum sources. In our study, therefore, plant species identity and spatial location were not confounded, and our results may constitute some of the best evidence for the roles of plant species identity and dispersal limitation in the determination of endophytic fungal community structure.

We found that there was a significant dispersal limitation in endophytic fungal communities of *Atriplex canescens*, accounting for 9% of community variation, and of *Bromus tectorum*, accounting for 17% of community variation. In the other four plant species, however, there

**Table 4. Results of pairwise PERMANOVAs among plant species and distances between centroids of leaf endophytic fungal communities for specific plant species.**

| | *Bromus tectorum* | *Chrysothamnus depressus* | *Artemisia tridentata* | *Elymus elymoides* | *Atriplex canescens* |
|---|---|---|---|---|---|
| *Bromus tectorum* | | | | | |
| *Chrysothamnus depressus* | $P < 0.001$<br>D = 0.349 (0.04) | | | | |
| *Artemisia tridentata* | $P < 0.001$<br>D = 0.362 (0.04) | $P = 0.299$<br>D = 0.236 (0.05) | | | |
| *Elymus elymoides* | $P = 0.367$<br>D = 0.221 (0.04) | $P < 0.001$<br>D = 0.338 (0.04) | $P < 0.001$<br>D = 0.337 (0.04) | | |
| *Atriplex canescens* | $P < 0.001$<br>D = 0.500 (0.05) | $P < 0.001$<br>D = 0.510 (0.05) | $P < 0.001$<br>D = 0.508 (0.05) | $P < 0.001$<br>D = 0.477 (0.06) | |
| *Alyssum alyssoides* | $P < 0.001$<br>D = 0.616 (0.04) | $P < 0.001$<br>D = 0.513 (0.03) | $P < 0.001$<br>D = 0.647 (0.03) | $P < 0.001$<br>D = 0.543 (0.05) | $P < 0.001$<br>D = 0.542 (0.03) |

*P* values are displayed in the first row. Distance to centroid of each plant species is displayed in the second row. Standard errors for distances, calculated by jackknife resampling, are given in parentheses.

was no significant dispersal limitation among the plots. The fact that for four plant species there was no significant dispersal limitation is, perhaps, not completely unexpected given the fact that the maximum distance between plots in this study was only 350 meters. We previously studied dispersal limitation of endophytic fungal communities of *Quercus gambelii*, another inhabitant of the eastern Great Basin [31]. In that study the maximum distance between sites was 15 km and dispersal limitation accounted for only between 3 and 8% of the variability in community structure. The surprising result is that in the current study the endophytic fungal communities of *Atriplex canescens* and *Bromus tectorum* leaves *did* exhibit a significant dispersal limitation across a maximum distance of only 350 meters. This suggests that, at least for these two species, the proximity of revegetated plants to a source of endophytic fungal inoculum may significantly influence the structure of endophytic fungal communities, even on a relatively small spatial scale. Proximity of revegetation plants to the inoculum source may thus be an important consideration when using this technology for revegetation purposes.

We also found that plant species identity accounted for 33% of the variation in endophytic fungal community structure. In other words, the six plant species possessed endophytic fungal communities that were quite different from each other. In some cases, the differences were caused by fungal OTUs that colonized some plant species but not others. Others have also found that the identity of the plant, either at the level of species [11,33] or genotype [34–37] influences the structure of endophytic fungal communities. Obviously if beneficial inoculant fungi had limited host breadth, inocula may have to be separately developed for different plant species. Nevertheless, there were 18 fungal taxa that were capable of colonizing leaves of all six plant species, although sometimes at markedly different frequencies. This indicates that among several of the endophytic fungal taxa in the eastern Great Basin, there is broad host plant compatibility. Therefore, it may be possible to develop a single beneficial inoculant fungus with broad host compatibility, which would simplify using inoculants to improve revegetation.

We conclude that if we implement endophytic fungal inoculation schemes to improve revegetation success, we must take into consideration both inoculum dispersal limitation and plant-fungus compatibility in order to achieve high levels of effectiveness and cost-efficiency. While plant species identity was more important than dispersal limitation in this study, the relative impacts of these two factors may depend on spatial scale; over larger spatial distances dispersal limitation is expected to increase in importance. Their relative impacts may also depend somewhat on year to year variation in average windspeed, and on the timing of plant establishment and leaf growth, which may relate to factors such as rainfall and temperature. One limitation of this study is that it was carried out during a single growing season and one might expect variation in the relative importance of dispersal limitation and species identity to vary by year.

Our results suggest a few additional hypotheses that warrant future testing. First, among the six plant species of our study, phylogenetic distance was a significant determinant of the structure of endophytic fungal communities, accounting for some 29% of total variability. As there is less phylogenetic distance among plant species within a family than among plant families, it was not surprising to find significant variation in the structure of endophytic fungal communities among plant families, and no significant variation among plant species within a plant family (either the Poaceae or the Asteraceae). A significant effect of plant phylogeny on the structure of endophytic fungal communities was also consistent with the fact that, for example, some fungal taxa were most frequently occurring in the Poaceae, or were most frequently occurring in the Poaceae and Asteraceae and less frequently occurring in the nonmycorrhizal species (*Atriplex canescens* and *Alyssum alyssoides*). The impact of plant phylogeny has not

been significant in every study. For example, Vincent et al. [33] suggested that tree species relatedness was not a significant factor determining the structure of endophytic fungal communities. Possibly plant phylogeny is important only at higher fungal taxonomic levels [58]. In any case, our study included only six plant species and thus offered only a limited ability to test the role of plant phylogeny. We feel, therefore, that the plant phylogeny hypothesis warrants proper testing in the future. The implication of this, however, may be a plant may serve as an inoculum source of a range of fungal taxa that are compatible with other members of the same family.

Second, our results also suggest the hypothesis that mycorrhizal status of plant species significantly influences the structure of endophytic fungal communities. The Amaranthaceae and Brassicaceae are both generally nonmycorrhizal or weakly mycorrhizal [59–62]. While their mycorrhizal status differs from that of the Asteraceae and Poaceae, which are both generally mycorrhizal, the Brassicaceae and Amaranthaceae are more closely related to the Asteraceae than Poaceae is to the Asteraceae. Yet the endophytic fungal communities of *Atriplex canescens* and *Alyssum alyssoides* were more dissimilar to those of the Asteraceae than those of the Poaceae were to those of the Asteraceae. This distinction between the nonmycorrhizal and the mycorrhizal plant species suggests that the ability of plants and *mycorrhizal* fungi to engage in the necessary molecular dialog to effect root colonization may be important in structuring foliar fungal communities. Our sample size was too small to test this hypothesis, but we feel that our result warrant further exploration of the mycorrhizal status hypothesis.

Third, some fungal taxa were most frequently occurring in *Atriplex canescens*, or most frequently occurring in *Alyssum alyssoides*. Among the fungal taxa that most frequently occurred in *Atriplex canescens* and *Alyssum alyssoides* were unknown species in the Pleosporales, and this order of fungi appeared to be unexpectedly diverse in *Atriplex canescens* and *Alyssum alyssoides* leaves. This suggests that some fungal lineages may have adapted to colonize nonmycorrhizal plant taxa and since adaptively radiated. That hypothesis may also warrant further testing.

Fourth, the endophytic fungal community of *Alyssum alyssoides* (Brassicaceae) was significantly different from that of *Atriplex* canescens (Amaranthaceae). While both families are largely nonmycorrhizal, they apparently utilize different mechanisms to limit mycorrhizal colonization [62], with the possible involvement of systemic mustard oils in the Brassicaceae but not in the Amaranthaceae [63]. Further experimentation into the mechanisms by which plants regulate mycorrhizal and endophytic fungi also appears to be warranted.

Finally, *Bromus tectorum* is a problematic invasive plant species in much of the arid western portion of the United States [64]. Its success as an invasive species may be partly determined by enhanced vigor associated with colonization by particular endophytic fungi [4,65]. Therefore, invasion of habitat by *Bromus tectorum* may be facilitated by the presence of sources of inoculum for mutualistic endophytic fungi [4]. As we learn more about the nature of interactions between specific host plants and specific endophytic fungi, it may be possible to manipulate endophytic fungi inoculum sources to disfavor *Bromus tectorum* while benefiting native plant species.

## Supporting information

**S1 Fig. OTU accumulation curves showing the increase in OTUs detected with increasing read depth.** All samples have been rarefied to an equal depth of 3,000 reads. Each line represents an individual sample. Curves were generated using the rarefy function in the Vegan package [51].
(TIF)

## Acknowledgments

We thank the anonymous reviewers for valuable comments on previous versions of this manuscript.

## Author Contributions

**Conceptualization:** Kevin D. Ricks, Roger T. Koide.

**Data curation:** Kevin D. Ricks.

**Formal analysis:** Kevin D. Ricks.

**Funding acquisition:** Roger T. Koide.

**Investigation:** Kevin D. Ricks, Roger T. Koide.

**Methodology:** Kevin D. Ricks, Roger T. Koide.

**Project administration:** Roger T. Koide.

**Resources:** Roger T. Koide.

**Supervision:** Roger T. Koide.

**Visualization:** Kevin D. Ricks.

**Writing – original draft:** Roger T. Koide.

**Writing – review & editing:** Kevin D. Ricks, Roger T. Koide.

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
