## [Decision Letter · Decision Letter 0]

24 Jun 2019

PONE-D-19-15145

The role of inoculum dispersal and plant species identity in the assembly of leaf endophytic fungal communities

PLOS ONE

Dear Dr. Koide,

Thank you for submitting your manuscript to PLOS ONE. After careful consideration, we feel that it has merit but does not fully meet PLOS ONE’s publication criteria as it currently stands. Therefore, we invite you to submit a revised version of the manuscript that addresses the points raised during the review process.

We would appreciate receiving your revised manuscript by Aug 08 2019 11:59PM. To enhance the reproducibility of your results, we recommend that if applicable you deposit your laboratory protocols in protocols.io, where a protocol can be assigned its own identifier (DOI) such that it can be cited independently in the future. For instructions see: http://journals.plos.org/plosone/s/submission-guidelines#loc-laboratory-protocols

We look forward to receiving your revised manuscript.

Kind regards,

Cheng Gao

Academic Editor

PLOS ONE

Journal Requirements:

2.

We note that you have indicated that data from this study are available upon request. PLOS only allows data to be available upon request if there are legal or ethical restrictions on sharing data publicly. For more information on unacceptable data access restrictions, please see http://journals.plos.org/plosone/s/data-availability#loc-unacceptable-data-access-restrictions.

Reviewers' comments:

Reviewer's Responses to Questions

**Comments to the Author**

1. Is the manuscript technically sound, and do the data support the conclusions?

Reviewer #1: Yes

Reviewer #2: Yes

2. Has the statistical analysis been performed appropriately and rigorously? 

Reviewer #1: Yes

Reviewer #2: Yes

3. Have the authors made all data underlying the findings in their manuscript fully available?

Reviewer #1: Yes

Reviewer #2: Yes

4. Is the manuscript presented in an intelligible fashion and written in standard English?

Reviewer #1: Yes

Reviewer #2: Yes

5. Review Comments to the Author

Reviewer #1: The authors explored the important effects of inoculum dispersal and plant species identity on community assembly of endophytic fungi. The study design is appropriate, as are the methods used, the paper is well-written and therefore I think it deserves publication in PLoS One.

Only three minor comments:

1. The letter “P” (P value) should be italic in Table 1, Table 4, Line 148, Line 164, Line 224, Line 232, Line 234.

2. “sp” should not be written in italic (Line 204-208, Line 218, Line259-274, Table 3)

3. In discussion, it would be appreciated if authors can cite of the major drawbacks encountered in this study and cite some of the future works.

Reviewer #2: This manuscript investigated the effects of both plant identity and fungal inoculum dispersal limitation on endophytic fungal community structure in leaves of six plant species in the eastern Great Basin of the USA, which are interesting. I have some minor comments and suggestions before publication:

Materials and methods:

Ln 95-102: You should give more information about the plots, for example, how did you establish the plots? and the distance among plots. In addition, it is unclear how many samples were actually sampled and sequenced. 18 replicated plots, and for each plot, a single sample for each of the six plant species, so 108 samples were collected and sampled? You should state clearly here.

Ln 106-107: How did you test the effectiveness of surface sterilization? Did you collect the sterile water used in the rinse and do the PCR or just culture on medium?

Ln 116: 5.8S Fun and ITS4 Fun

Ln 128-131: Why did you rarefy the samples prior to all sequence analyses? I saw the Figure S1, the sequence depth is 3000. So this step is the same with or different from the rarefaction you mentioned in Ln 141-142?

Ln136-Ln139: You amplified the ITS2 region using the 5.8S Fun and ITS4 Fun. Generally, some studies will extracted ITS before the OTU cluster, Why did not you do that? In addition, did you check the chimera? What was the proportion of chimeric sequences?

Ln 140: What was the proportion of your ITS sequences belonged to the host in each sample?

Have authors deposited sequences to a public repository? You should state in the manuscript.

Results:

Ln 174: How many raw data were obtained for each sample? And how about the reads for each sample before rarefaction?

L173-176: Maybe it is better to add some description about the fungal taxonomic composition in the first paragraph. You should provide a supporting information about your fungal identification.

Ln 215: Please cite the "Table 3" behind "among plant families"

Table 3: I think this table does not show the information visually. Maybe you can colored the cell of the table according to the frequencies of occurrence, like a heatmap.

Discussion:

Ln 317-318: Are there any other publications describing the 18 fungal taxa colonizing leaves of other plant species? And are there publications showing these fungal taxa contribute to the host health and development? Maybe you can add some discussion about this.

6. PLOS authors have the option to publish the peer review history of their article (what does this mean?). If published, this will include your full peer review and any attached files.

Reviewer #1: No

Reviewer #2: No

---

## [Author Response · Author response to Decision Letter 0]

27 Jun 2019

2.

We note that you have indicated that data from this study are available upon request. PLOS only allows data to be available upon request if there are legal or ethical restrictions on sharing data publicly. For more information on unacceptable data access restrictions, please see http://journals.plos.org/plosone/s/data-availability#loc-unacceptable-data-access-restrictions.

Our sequence data were submitted to the sequence read archive at NCBI. We now include the accession number.

Our sequence data were submitted to the sequence read archive at NCBI. We now include the accession number.

Reviewers' comments:

Reviewer's Responses to Questions

Comments to the Author

1. Is the manuscript technically sound, and do the data support the conclusions?

Reviewer #1: Yes

Reviewer #2: Yes

2. Has the statistical analysis been performed appropriately and rigorously?

Reviewer #1: Yes

Reviewer #2: Yes

3. Have the authors made all data underlying the findings in their manuscript fully available?

Reviewer #1: Yes

Reviewer #2: Yes

4. Is the manuscript presented in an intelligible fashion and written in standard English?

Reviewer #1: Yes

Reviewer #2: Yes

5. Review Comments to the Author

Reviewer #1: The authors explored the important effects of inoculum dispersal and plant species identity on community assembly of endophytic fungi. The study design is appropriate, as are the methods used, the paper is well-written and therefore I think it deserves publication in PLoS One.

Only three minor comments:

1. The letter “P” (P value) should be italic in Table 1, Table 4, Line 148, Line 164, Line 224, Line 232, Line 234.

This has been corrected.

2. “sp” should not be written in italic (Line 204-208, Line 218, Line259-274, Table 3)

This has been corrected.

3. In discussion, it would be appreciated if authors can cite of the major drawbacks encountered in this study and cite some of the future works.

Some limitations of the study are now included in the fifth paragraph of the discussion. More than half of the discussion is devoted to future work in the form of hypotheses that we feel merit testing, and the reasons why we feel they should be tested.

Reviewer #2: This manuscript investigated the effects of both plant identity and fungal inoculum dispersal limitation on endophytic fungal community structure in leaves of six plant species in the eastern Great Basin of the USA, which are interesting. I have some minor comments and suggestions before publication:

Materials and methods:

Ln 95-102: You should give more information about the plots, for example, how did you establish the plots? and the distance among plots.

These are now given in more detail in the second paragraph of the materials and methods.

In addition, it is unclear how many samples were actually sampled and sequenced. 18 replicated plots, and for each plot, a single sample for each of the six plant species, so 108 samples were collected and sampled? You should state clearly here.

That is correct. This is now specified in the second paragraph of the materials and methods.

Ln 106-107: How did you test the effectiveness of surface sterilization? Did you collect the sterile water used in the rinse and do the PCR or just culture on medium?

We tested the effectiveness of the method by determining the extent to which it and other methods were capable of reducing total fungal species richness. The sodium hypochlorite method that we used was significantly more effective than ethanol, which appears to be a popular method used by others in this field. Thank you for the suggestion about testing the sterile water rinse. However, the sterile water rinse would not have been an acceptable way to test the efficacy of the method because fungi or fungal DNA could still be adhering to the leaf surface and not appear in the rinse.

Ln 116: 5.8S Fun and ITS4 Fun

Corrected.

Ln 128-131: Why did you rarefy the samples prior to all sequence analyses?

As we specified in the text, “This made samples comparable despite the potential for different original sequencing depths.” Rarefaction is a common practice in this kind of study to ensure comparability when calculating such things as diversity, evenness, etc. because it gives each sample the same sampling depth.

I saw the Figure S1, the sequence depth is 3000. So this step is the same with or different from the rarefaction you mentioned in Ln 141-142?

Yes, this is the same and it is now more clearly specified.

Ln136-Ln139: You amplified the ITS2 region using the 5.8S Fun and ITS4 Fun. Generally, some studies will extracted ITS before the OTU cluster, Why did not you do that?

Unfortunately, we do not understand this comment

In addition, did you check the chimera? What was the proportion of chimeric sequences?

Yes. Using the UCHIME function, 0.8% of reads were identified as chimeras, and these were removed. We now include this information in the 6th paragraph of the materials and methods.

Ln 140: What was the proportion of your ITS sequences belonged to the host in each sample?

6% of the total reads belonged to the host. This information is now included in the 6th paragraph of the materials and methods.

Have authors deposited sequences to a public repository? You should state in the manuscript.

Yes. FASTQ files are deposited in the Sequence Read Archive (SRA) at NCBI (accession number PRJNA518913). This information is now included in the 6th paragraph of the materials and methods.

Results:

Ln 174: How many raw data were obtained for each sample? And how about the reads for each sample before rarefaction?

There was an average of 12,816 per sample before filtering. After filtering and prior to rarefaction there was an average of 7167 per sample. This information is now given in the first paragraph of the Results section.

L173-176: Maybe it is better to add some description about the fungal taxonomic composition in the first paragraph. You should provide a supporting information about your fungal identification.

This description is now given in the form of a new figure (Fig 3).

Ln 215: Please cite the "Table 3" behind "among plant families"

This has been corrected.

Table 3: I think this table does not show the information visually. Maybe you can colored the cell of the table according to the frequencies of occurrence, like a heatmap.

It may be possible to turn this table into a figure as the reviewer suggests. However, we felt that more information could be conveyed by presenting the actual frequencies rather than colors coded to frequency. Our goal was to simultaneously show the frequencies, and color-code the columns so plant families were clear.

Discussion:

Ln 317-318: Are there any other publications describing the 18 fungal taxa colonizing leaves of other plant species? And are there publications showing these fungal taxa contribute to the host health and development? Maybe you can add some discussion about this.

Of the 18 fungal taxa that were found in all six species, only 7 were identifiable to species. Even in these 7 cases, it is not clear whether knowing if they colonize other plant species and influence their health in any way is useful because the effects of endophytic fungal colonization is context dependent (dependent on host species and environmental conditions) and on the specific isolate of the fungal species. Moreover, the main point of our findings is not to suggest that any of these 18 fungal species could be used to promote revegetation of the great basin plant species that we studied. The point of our study was to determine the extent to which plant identity and dispersal limitation generally influence endophytic fungal community structure. That knowledge will influence how we use inoculant fungi in the future, not necessarily which fungi we choose for that purpose.

---

## [Decision Letter · Decision Letter 1]

3 Jul 2019

The role of inoculum dispersal and plant species identity in the assembly of leaf endophytic fungal communities

PONE-D-19-15145R1

Dear Dr. Koide,

We are pleased to inform you that your manuscript has been judged scientifically suitable for publication and will be formally accepted for publication once it complies with all outstanding technical requirements.

With kind regards,

Cheng Gao

Academic Editor

PLOS ONE

Additional Editor Comments (optional):

Reviewers' comments:

Reviewer's Responses to Questions

**Comments to the Author**

1. If the authors have adequately addressed your comments raised in a previous round of review and you feel that this manuscript is now acceptable for publication, you may indicate that here to bypass the “Comments to the Author” section, enter your conflict of interest statement in the “Confidential to Editor” section, and submit your "Accept" recommendation.

Reviewer #1: All comments have been addressed

Reviewer #2: All comments have been addressed

2. Is the manuscript technically sound, and do the data support the conclusions?

Reviewer #1: Yes

Reviewer #2: Yes

3. Has the statistical analysis been performed appropriately and rigorously? 

Reviewer #1: Yes

Reviewer #2: Yes

4. Have the authors made all data underlying the findings in their manuscript fully available?

Reviewer #1: Yes

Reviewer #2: Yes

5. Is the manuscript presented in an intelligible fashion and written in standard English?

Reviewer #1: Yes

Reviewer #2: Yes

6. Review Comments to the Author

Reviewer #1: (No Response)

Reviewer #2: (No Response)

7. PLOS authors have the option to publish the peer review history of their article (what does this mean?). If published, this will include your full peer review and any attached files.

Reviewer #1: No

Reviewer #2: No

---

## [Editor Report · Acceptance letter]

9 Jul 2019

PONE-D-19-15145R1

The role of inoculum dispersal and plant species identity in the assembly of leaf endophytic fungal communities

Dear Dr. Koide:

I am pleased to inform you that your manuscript has been deemed suitable for publication in PLOS ONE. Congratulations! Your manuscript is now with our production department.

With kind regards,

on behalf of

Dr. Cheng Gao

Academic Editor

PLOS ONE